# Trends and spatial distribution of animal bites and vaccination status among victims and the animal population, Uganda: A veterinary surveillance system analysis, 2013–2017

Fred Monje[1]*, Daniel Kadobera[1], Deo Birungi Ndumu[2], Lilian Bulage[1], Alex Riolexus Ario[1]

1 Uganda Public Health Fellowship Program, Ministry of Health, Kampala, Uganda, 2 National Animal Disease Diagnostics and Epidemiology Centre, Ministry of Agriculture, Animal Industry and Fisheries, Entebbe, Uganda

* fredmonje@musph.ac.ug

**Data Availability Statement:** The datasets upon which our findings are based are owned by the government of Uganda. To access this dataset, we

## Abstract

Rabies is a vaccine-preventable fatal zoonotic disease. Uganda, through the veterinary surveillance system at National Animal Disease Diagnostics and Epidemiology Centre (NADDEC), captures animal bites (a proxy for rabies) on a monthly basis from districts. We established trends of incidence of animal bites and corresponding post-exposure prophylactic anti-rabies vaccination in humans (PEP), associated mortality rates in humans, spatial distribution of animal bites, and pets vaccinated during 2013–2017. We reviewed rabies surveillance data at NADDEC from 2013–2017. The surveillance system captures persons reporting bites by a suspected rabid dog/cat/wild animal, human deaths due to suspected rabies, humans vaccinated against rabies, and pets vaccinated. Number of total pets was obtained from the Uganda Bureau of Statistics. We computed incidence of animal bites and corresponding PEP in humans, and analyzed overall trends, 2013–2017. We also examined human mortality rates and spatial distribution of animal bites/rabies and pets vaccinated against rabies. We identified 8,240 persons reporting animal bites in Uganda during 2013–2017; overall incidence of 25 bites/ 100,000population. The incidence significantly decreased from 9.2/100,000 in 2013 to 1.3/100,000 in 2017 (OR = 0.62, p = 0.0046). Of the 8,240 persons with animal bites, 6,799 (82.5%) received PEP, decreasing from 94% in 2013 to 71% in 2017 (OR = 0.65, p<0.001). Among 1441 victims, who reportedly never received PEP, 156 (11%) died. Western region had a higher incidence of animal bites (37/100,000) compared to other regions. Only 5.6% (124,555/2,240,000) of all pets in Uganda were vaccinated. There was a decline in the reporting rate (percentage of annual district veterinary surveillance reports submitted monthly to Commissioner Animal Health by districts) of animal bites. While reported animal bites by districts decreased in Uganda, so did PEP among humans. Very few pets received anti-rabies vaccine. Evaluation of barriers to complete reporting may facilitate interventions to enhance surveillance quality. We recommended improved vaccination of pets against rabies, and immediate administration of exposed humans with PEP.

can contact the rabies focal person, Dr. Moses Mwanja, Email: drmwanja@gmail.com.

**Funding:** This project was supported by the President's Emergency Plan for AIDS Relief (PEPFAR) through the US Centers for Disease Control and Prevention Cooperative Agreement number GH001353–01 and through Makerere University School of Public Health to the Uganda Public Health Fellowship Program, MoH. The staff of the funding body provided technical guidance in the design of the study, ethical clearance and collection, analysis, and interpretation of data and in writing the manuscript.

**Competing interests:** The authors have declared that no competing interests exist.

## Author summary

Rabies is a deadly viral disease, that is transmitted mainly by dog bites. Globally at least 59,000 deaths are reported to occur annually- mostly in Sub-Saharan Africa and Asia. However, rabies can be prevented through vaccination of pets (dogs and cats) and administration of rabies vaccine in humans exposed to rabies. In our study we reviewed secondary data of animal bites and rabies captured at the National Animal disease diagnostic epidemiology centre in Entebbe for the period 2013–2017. We found that of 1441 animal bite victims who never received rabies vaccine, only 156 (11%) died hence need for immediate administration of exposed humans with rabies and sensitization of the public about the consequences of animal bites and need for urgent health care. There was a decline in the reporting rate of animal bites during the study period suggesting that evaluation of the barriers to complete reporting may facilitate interventions to enhance surveillance quality. Less than 10% of the pets in the Uganda were vaccinated against rabies hence need for improved vaccination of pets against rabies through appropriated legislation.

## Introduction

Rabies is a fatal viral vaccine-preventable zoonotic disease that can infect warm-blooded animals [1–3]. Worldwide, canine rabies causes an estimated 59,000 human deaths and 8.6 billion USD (95% CIs: 2.9–21.5 billion) in economic losses annually [3]. Rabies is transmitted primarily through bites from an infected rabid animal; however, it can also be transmitted from licks or scratches from an infected rabid animal or, rarely, through transplantation of tissues or organs from an infected individual [4–6]. Rabies occurs on all continents except Antarctica, with most cases reported in Africa and Asia [7, 8]. Nearly all cases of human rabies are due to bites from infected dogs [7]. In dogs, the incubation period of rabies is 10 days to 6 months, with most cases manifesting signs between 2 weeks to 3 months after exposure [9]. In humans, the average incubation period of rabies is 20–60 days, though it can last up to several years [6, 10].

Prevention of rabies in animals is primarily achieved through vaccination. Indeed, in developed countries, mass canine vaccination coupled with oral vaccination in wildlife have greatly contributed to the elimination of rabies in canines, and consequently a reduction in human rabies [7]. In humans, rabies prevention typically occurs through rabies Post-Exposure Prophylaxis (rPEP) in the form of a rabies vaccine. This vaccine should be administered to victims of bites from suspected rabid animals as soon as possible, and continued while the animal is being observed for 10–14 days or pending the results of laboratory tests [6]. However, rPEP requires multiple doses, is not always available, and must be administered in a timely manner to be effective. The most cost-effective method of controlling rabies is to prioritize canine vaccination, rather than using reactive rPEP in humans [7].

In Uganda, Ministry of Agriculture, Animal Industry and Fisheries (MAAIF) procures anti-rabies vaccine annually to control rabies in animals, though in limited doses [11]. This vaccine is provided to districts based on the magnitude of reported animal bites, the dog population in the district, and confirmed rabies cases in pets. Historically, MAAIF has allocated about 2,000 doses of anti-rabies vaccine for pets to each of the districts in Uganda affected by rabies. This is generally insufficient for the estimated pet populations in each of the districts. While people are able to obtain human rabies vaccine (rPEP) from private providers (such as

pharmacies, private health centres) following exposure to a suspected rabid animal, the cost ($8–12 USD per dose) is prohibitive for most [12].

Among humans, an animal bite is treated as being from a rabid animal, until proved otherwise. Surveillance data of animal bites among humans provides vital information to guide resource allocation in the control and prevention of rabies [13]. An analysis of human rabies surveillance data from the Epidemiology and Surveillance Division (ESD) of the Ministry of Health in 2001–2015 in Uganda revealed 208,720 animal bites, with 486 suspected human rabies deaths [14]. However, Uganda also captures rabies-related data in a veterinary surveillance system, at National Animal Disease Diagnostic Epidemiology Centre (NADDEC) to aid in animal disease monitoring and surveillance. Beyond capturing animal bites and suspected human rabies deaths, NADDEC also captures the number of pets vaccinated against rabies (including by private providers) and conducts confirmatory tests of suspected rabid animals. We used data from NADDEC to describe trends in the incidence of animal bites and corresponding rPEP in humans, mortality associated with animal bites in humans, spatial distribution of animal bites, and pets vaccinated in Uganda from 2013–2017.

## Methods

### Ethics statement

We got verbal permission to conduct this investigation from the NADDEC, of the Ministry of Agriculture, Animal Industry and Fisheries. Additionally, CDC determined that this investigation was a public health emergency activity whose primary intent was to control rabies at the source (animals) such that exposure and transmission to humans is curtailed, and therefore it was classified as not research.

### Study setting and design

We conducted the study using national surveillance data from National Animal Disease Diagnostics and Epidemiology Centre (NADDEC) of the Ministry of Agriculture, Animal Industry and Fisheries. When a bite from a suspected rabid animal is reported in the communities, Uganda guidelines require that it must be immediately reported as soon as possible to the nearest veterinary officer or animal husbandry officer at the sub-county headquarters. The officer visits the scene of the incident, assesses the circumstances under which the victim was bitten and the vaccination history of the dog/cat at the time of the incident, and advises whether or not the suspected rabid animal (usually dog/cat) should be killed and its head packaged for shipment to NADDEC for laboratory testing. Meanwhile, the veterinary officer/animal husbandry officer writes a referral letter for the victim to receive rPEP at the nearest health facility (usually Health centre IV, Hospital, or Referral Hospital). At the health facility, there are usually limited doses of human rabies vaccine and rabies immunoglobin. The standard rPEP regimen against rabies in Uganda is administered on days 0, 3, 7, 14, and 28 into the deltoid muscle. However, there is no data on the total number of doses received. If the veterinary/ AH officer issues a referral letter for rPEP to the bite victim, who fails to attend the health facility, or there is no vaccine/RIG available on arrival at the facility, then this is recorded as "did not receive rPEP". A monthly sub-county report with all the animal bites is compiled by the officer and shared with the District Veterinary Officer (DVO), for compilation and entry into a standard veterinary surveillance form. The DVO then submits a district monthly veterinary surveillance report to the Commissioner for Animal Health (CAH), and NADDEC for compilation. In case any human death occurs weeks or months after the bite, this data is gathered and reported through the health facility system (health management information system). Additionally, there is a system of reporting suspect human rabies cases by health facility back

to the veterinary or AH officer who also reports it through the DVO to CAH and NADDEC. The human rabies data for suspect human cases are based solely on clinical presentation. The findings from veterinary disease data are intended to be shared with the districts through annual DVO meetings [15].

We carried out a retrospective descriptive study involving review of rabies data captured in standard veterinary surveillance forms at NADDEC during 2013–2017 from all districts in Uganda. We extracted rabies-specific variables from standard veterinary surveillance forms at NADDEC. For example: date (month and year), district name, number of suspected cases in animals, number of bites by suspected animals, number of cases in humans (deaths and emergency vaccinations), vaccinations in dogs and cats, and number of pets destroyed. We excluded all records that had had missing variables such as district name and date. To understand more about the completeness of reporting, we also captured information about the number of monthly surveillance reports expected in a year at NADDEC, and number of reports actually received.

To get information about confirmed rabies cases in animals during 2013–2017, we reviewed NADDEC laboratory records for rabies testing for suspected animals. Rabies diagnosis was conducted on animal brain tissue using the direct fluorescent antibody test (dFAT, Onderstepoort Veterinary Institute, Pretoria, South Africa) at NADDEC [16].

### Data management and analysis

We extracted monthly epidemiological rabies data from standard veterinary surveillance forms at NADDEC into Microsoft Excel and cleaned it before analysis. During, analysis we computed proportions, percentages and rates. We calculated annual and overall incidence of animal bites by using number of new cases as a numerator and total human population at risk as a denominator during 2013–2017. We drew maps using Quantum Geographic information system (QGIS) to demonstrate geographical distribution of animal bites by district. From the NADDEC data, we computed the annual trends in rPEP and mortality rates in humans associated with animal bites in the period 2013–2017. We also computed the proportion of pets vaccinated against rabies during the period 2013–2017, using pet population data obtained from the Uganda Bureau of Statistics from a 2008 census [17]. To do a trends analysis, we used logistic regression in Epiifo version 7.2.2.6.

### Results

We identified 8,240 reports of animal bites in Uganda during the study period, with an overall incidence of 25 animal bites per 100,000 population. The annual incidence significantly decreased from 9.2/100,000 in 2013 to 1.3/100,000 in 2017 (OR = 0.62, p = 0.0046) (Fig 1). Animal bites were reported in almost all districts during 2013-2017in Uganda (Fig 2). Western region had the highest incidence of animal bites (37/100,000), followed by Northern region (22/100,000), Central region (17/100,000), and Eastern region (17/100,000) (Fig 3). These differences were largely driven by differences in reported bites during 2013 and 2014; reported incidences were much lower among all regions during 2015–2017 (Fig 3). Laboratory data indicated that 36% (28/77) of the brain samples from suspected rabid animals tested positive for rabies during the study period.

Of the 8,240 human animal bite victims, 6,799 (82.5%) reportedly obtained rPEP. The number of doses of rPEP initiated was not recorded- instead what was captured was whether rPEP was initiated or not. The percentage obtaining rPEP after the bite decreased from 94% in 2013 to 71% in 2017 (OR = 0.65; p<0.001) (Fig 1). Among all human animal bite victims, 156 (1.9%) died. The annual reported mortality rates among bitten humans decreased from 3.0%

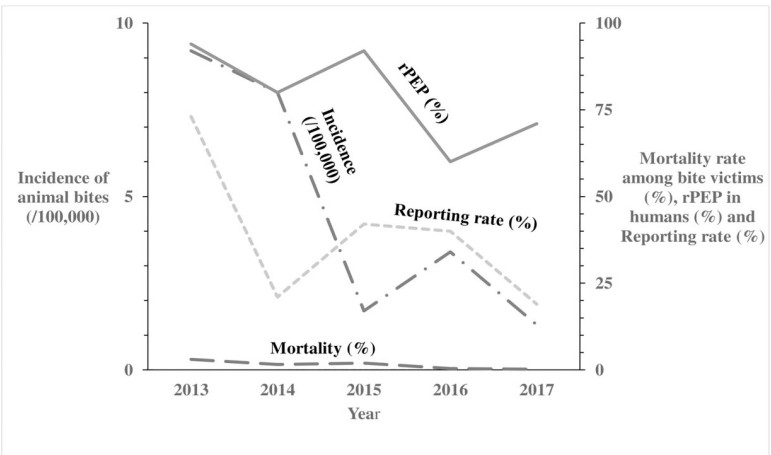

**Fig 1. Trends of incidence of animal bites, anti-rabies vaccinations in humans, corresponding mortality rates and reporting rates in Uganda: 2013–2017.** The Long Dash Dot line represents the incidence of animal bites (/100,000) in humans. The Solid line represents the anti-rabies vaccinations in humans. The Long Dash line represents the mortality rates in humans following a bite from a suspected rabid animal. The Square Dot line represents the reporting rates per year.

in 2013 to 0.21% in 2017 (OR = 0.58, p = 0.08) (Fig 1). Of the 2,221,620 pets, 6,997 (0.31%) were reported destroyed by the communities following infliction of bite injuries to humans; the proportion reported destroyed by the communities rose from 62% in 2013 to 69% in 2017 (OR = 1.2; p = 0.095) (Fig 4). During the study period, 124,555 rabies vaccines were provided for the estimated 2,221,620 pets (1,580, 930 dogs and 640,690 cats) in Uganda, resulting in a maximum of 5.6% of pets vaccinated. Of the 6,576 monthly veterinary reports expected from the districts, only 2,517 (38%) overall were received during the study period. The percentage of expected reports received decreased from 73% in 2013 to 19% in 2017 (OR = 0.67, P<0.001) (Fig 1).

## Discussion

We describe veterinary surveillance data for suspected rabies in Uganda, which shows changes in the regularity of district-level reporting of data and bite-associated human mortality in Uganda during 2013–2017. In addition, the proportion of bitten persons receiving rPEP decreased over time. Most of the animal samples tested did not test positive for rabies.

Although the data in this report appear promising for rabies reduction in Uganda, reporting rates (percentage of annual district veterinary surveillance reports submitted monthly to Commissioner Animal Health by districts) over the study period decreased greatly. During 2013 and 2014 in Uganda, active animal disease surveillance was being supported by donors, including the Pan African Control of Epizootics (PACE) and others [18]. This enabled district veterinary officers to physically bring hard-copy monthly reports to CAH, and hence to NAD-DEC, which improved the completeness of reporting and likely provided a more accurate picture of each of the metrics included in NADDEC rabies reporting. This support ended in 2015, which drastically affected not only rabies surveillance and reporting, but also other district-level animal disease reporting in Uganda. Other studies have also shown that passive surveillance for rabies results in underreporting [19], particularly in developing countries [3, 20]. Analysis of ESD rabies surveillance data from 2001–2015 found that animal bites were increasing in Uganda [14], and it is reasonable to assume that complete reporting in NADDEC might have shown a similar trend. Perhaps the use of eHealth and mobile technology could improve

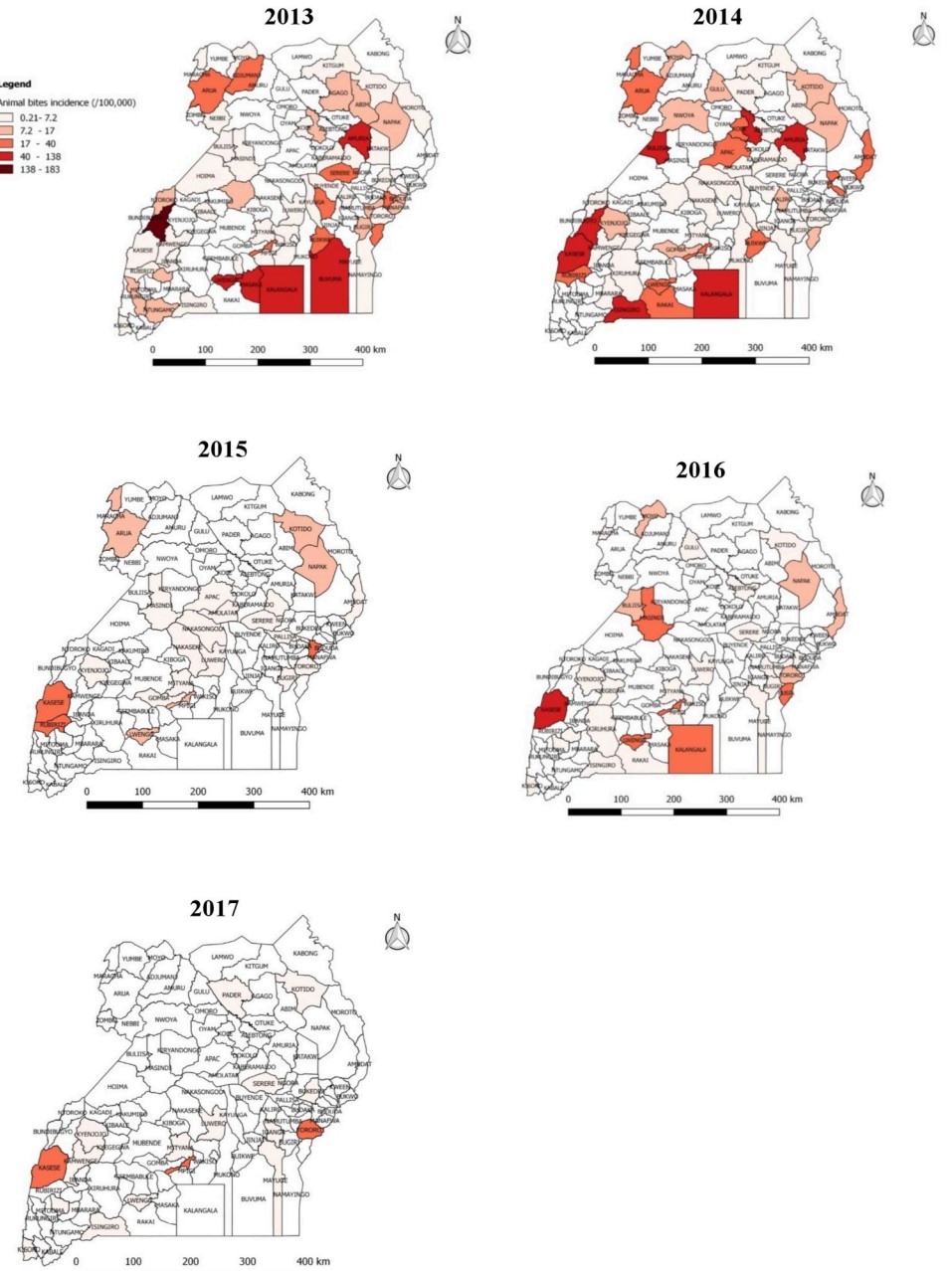

**Fig 2. Reported incidence (/100,000 population) of animal bites by district in Uganda, 2013–2017.** The variation in incidence of animal bites (/100,000) per district from 0.21–183.

the efficiency and timeliness of case reporting as has been demonstrated elsewhere without increasing cost [21, 22]. A full evaluation of both rabies surveillance systems and the challenges to complete reporting is important to enable the system to contribute to its fullest extent in Uganda.

Regardless, findings from this study provide important information for Uganda. Vaccinating pets is one of the most important ways to prevent rabies infections among both animals and humans [7, 23]. However, there are clear gaps in pet vaccination in Uganda; fewer than

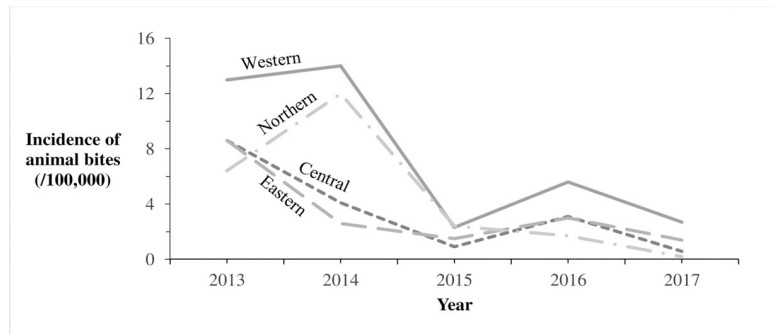

**Fig 3. Reported incidence of animal bites (/100,000) by region in Uganda: 2013–2017.** The Solid line represents incidence of animal bites (/100,000) in Western region. The Long Dash Dot line represents incidence of animal bites (/100,000) in Northern region. The Square Dot line represents incidence of animal bites (/100,000) in Central region. The Long Dash line represents incidence of animal bites (/100,000) in Eastern region.

one in ten pets were vaccinated in our analysis. This may be due in part to the limited quantities of anti-rabies vaccine procured by MAAIF for the districts [11]. However, a review of other studies highlights that most dogs are accessible for vaccination [24]. Furthermore, a sufficient proportion of dogs in most African communities are amenable to handling for parenteral vaccination [24]. One study indicated that the total dog population in Uganda may be less than previously estimated, although their findings were based on modelling using a small fraction of the entire population [25]. Though a relatively small fraction of regions were sampled in this study, the methods appear statistically sound and the power of the sample size adequate, which would have enabled valid estimation of the national population. A detailed Livestock census conducted by the Uganda bureau of Statistics in collaboration with the Ministry of Agriculture Animal industry and Fisheries estimated that the dog population was 1,580, 930 and cat population was 640, 690[17]. The initial estimate of the study at 1.3 million dogs is not hugely dissimilar to that of the livestock census at 1.6 million dogs. In view of the differences from these two estimates, we shall consider the Livestock census the more valid option for purposes of this study. Ultimately whether the total dog population is estimated at 0.7 million, 1.3 million or 1.6 million, the conclusion remains the same; dog vaccination in Uganda is inadequate and a systematic national dog vaccination campaign is needed.

A study in rural Uganda indicated that over 80% of dogs could be vaccinated in a pilot campaign through Static Point vaccination [26]. Another study pointed out that rabies control can be achieved with sufficient vaccination rates: a study modelling the minimum dog vaccination

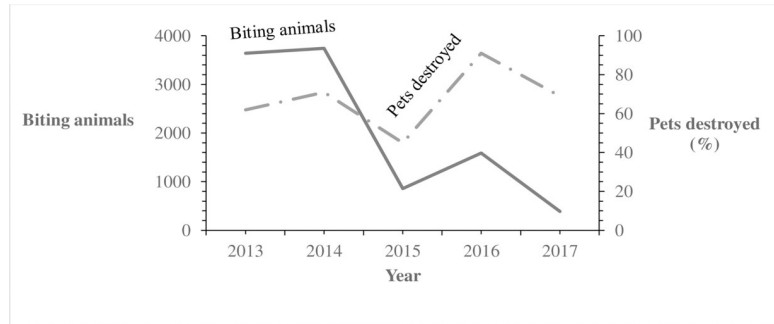

**Fig 4. Reported trends of pets destroyed (%) among biting animals in Uganda: 2013–2017.** The Solid line represents biting animals. The Long Dash Dot line represents pets destroyed.

coverage required for interruption of transmission of rabies in humans in Ndjamena, Chad showed that only 71% of dogs would require vaccination [27]. Mass oral rabies vaccination for free-ranging dogs (OVD), which has been researched and promoted by WHO since 1988, should also be a part of the rabies control strategy [28].

While the total numbers of biting animals reported declined, the proportion of biting animals destroyed appeared to increase slightly throughout the study period. However, not all biting pets were destroyed. The accuracy of these data is unclear. Some biting pets might have escaped before the community had an opportunity to kill them; this is a common occurrence [29, 30]. The community might also have destroyed the pet but not reported it to the DVO, as has been reported to occur elsewhere [12]. A follow up evaluation of knowledge, attitudes and practices around animal bites in the communities may be necessary to understand this issue more fully.

Most persons bitten by pets in our study received rPEP. In Uganda, rPEP is provided by government and can easily be found at high-level health centers and hospitals. However, the proportion of persons receiving rPEP in Uganda after a bite declined over the study period. The reasons for this are unclear but may relate to changes in availability at government health centers. In some areas, although rPEP is supposed to be provided free of charge, patients may be charged regardless, which can impact uptake. At least one study has shown that the costs of rPEP are typically underestimated [12]. Alternately, changes in healthcare-seeking behaviors may have occurred that resulted in the decline; however, further qualitative investigation would be needed to identify these. In Haiti, methods of Integrated Bite Case Management (IBCM) have shown considerable benefits in improving the investigation of suspect rabies cases [31, 32]. Probably methods of IBCM could also be of benefit in Uganda to increase compliance with rPEP in cases of highest risk.

Furthermore, our findings indicated that approximately 1/3 of animals tested had rabies. The heads that were sent for testing likely reflect animals that had a higher pre-test probability of having rabies, thus this proportion is likely to be an overestimate of overall rabies infection among animals biting humans. In addition, among persons who did not receive rPEP, few died, suggesting that most of the animals biting might not have rabies. This is consistent with other studies that have shown that most biting animals do not have rabies [32]. However, it is also possible that some animal bite victims accessed rPEP from private health facilities and were not captured in the rabies surveillance system in the veterinary sector. It is also possible that deaths due to suspected rabies could have been missed in surveillance, as rabies has a varied incubation period (three weeks to several years) and may not have been properly recognized and reported [6, 10]. A more detailed investigation and follow-up of persons bitten by animals, as well as active surveillance to track rPEP, could inform this issue.

Although animal bites were reported throughout Uganda, Western region had the highest incidence of animal bites during the study period. This may be due to its proximity to the Democratic Republic of Congo (DRC), which reported alarming numbers of people bitten by rabid dogs in 2013 [33, 34]. The thick forests of eastern DRC that border Uganda may shelter reservoirs of rabies such as jackals, and serve as sources of cross-border infections [34]. In rabies elimination efforts in Uganda, this area may need special attention.

## Limitations

We were unable to link individual rPEP results to outcomes, preventing us from knowing if persons who died did or did not receive rPEP. We also did not have data about their causes of death, making it uncertain if they died from rabies. Beyond this, we did not have recent data about the number of pets in Uganda, nor about changes over time, and could only access a

single value for pet numbers from the year 2008. The increase in human population in Uganda almost certainly indicates an increase in pets as well, which would have led to overestimations of vaccination rates. In addition, there was substantial underreporting, which almost certainly led to an underestimation of the magnitude of the animal bites and hence rabies in our study.

## Conclusions and recommendations

Animal bites decreased in Uganda, with western Uganda having the highest bite rate. Rabies PEP receipt among bite victims decreased over time, and overall, very few pets received anti-rabies vaccine nationwide. There was a decline in the reporting rate (percentage of annual district veterinary surveillance reports submitted monthly to Commissioner Animal Health by districts) during 2013–2017. Evaluation of barriers to complete reporting may facilitate interventions to enhance surveillance quality. We recommended improved vaccination of pets against rabies through legislation, immediate administration of exposed humans to post-exposure anti-rabies vaccine, and sensitization of the public about the consequences of animal bites and need for urgent health care.

## Acknowledgments

The authors are indebted to the Uganda's National Animal Disease Diagnostic Epidemiology Centre (NADDEC) of the Ministry of Agriculture, Animal Industry and Fisheries (MAAIF), for providing access to the rabies surveillance data and laboratory results of animal samples. We also appreciate Uganda Public Health Fellowship for the technical support during the design, analysis, and interpretation of this study. We thank PHFP cohort 2018 Fellows for the technical support during the execution of this study. The views and opinions expressed in this article are those of the authors and do not necessarily represent the official position of the US Centers for Disease Control and Prevention, the Department of Health and Human Services, Makerere University School of Public Health, or the MoH.

## Author Contributions

**Conceptualization:** Fred Monje, Daniel Kadobera, Deo Birungi Ndumu, Alex Riolexus Ario.

**Data curation:** Fred Monje.

**Formal analysis:** Fred Monje, Lilian Bulage, Alex Riolexus Ario.

**Investigation:** Fred Monje, Deo Birungi Ndumu, Alex Riolexus Ario.

**Methodology:** Fred Monje, Daniel Kadobera, Deo Birungi Ndumu, Lilian Bulage, Alex Riolexus Ario.

**Project administration:** Fred Monje.

**Supervision:** Daniel Kadobera.

**Validation:** Fred Monje, Lilian Bulage, Alex Riolexus Ario.

**Visualization:** Fred Monje, Lilian Bulage, Alex Riolexus Ario.

**Writing – original draft:** Fred Monje, Daniel Kadobera, Deo Birungi Ndumu, Lilian Bulage.

**Writing – review & editing:** Fred Monje, Daniel Kadobera, Deo Birungi Ndumu, Lilian Bulage, Alex Riolexus Ario.

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
