## [Decision Letter · Decision Letter 0]

5 Feb 2020

Dear Dr Monje,

Thank you very much for submitting your manuscript "Trends and spatial distribution of animal bites and vaccination status among victims and the animal population, Uganda: A veterinary surveillance system analysis, 2013 – 2017" for consideration at PLOS Neglected Tropical Diseases. As with all papers reviewed by the journal, your manuscript was reviewed by members of the editorial board and by several independent reviewers. In light of the reviews (below this email), we would like to invite the resubmission of a significantly-revised version that takes into account the reviewers' comments. 

The authors report on a timely and important topic that will be of interest to all authorities and others involved in neglected tropical diseases and in particular the global tri-partite agenda to end human rabies deaths mediated by dogs by 2030. Reports such as this one are important to call attention to the diagnostic, infrastructure and/or reporting gaps which could compromise efforts towards the global goal. Nevertheless, the methods of the current study are incompletely described and referenced, which call the validity of the results and conclusions into question. Please provide the necessary level of detail in the methods so that the study may be reproducible by others. The Discussion should also address the limitations of the data analysis more clearly and acknowledge how this study advances knowledge in the context of other recent and relevant background literature highlighted by the reviewers.

We cannot make any decision about publication until we have seen the revised manuscript and your response to the reviewers' comments. Your revised manuscript is also likely to be sent to reviewers for further evaluation.

Sincerely,

Amy T. Gilbert

Guest Editor

Sergio Recuenco

Deputy Editor

The authors report on a timely and important topic that will be of interest to all authorities and others involved in neglected tropical diseases and in particular the global tri-partite agenda to end human rabies deaths mediated by dogs by 2030. Reports such as this one are important to call attention to the diagnostic, infrastructure and/or reporting gaps which could compromise efforts towards the global goal. Nevertheless, the methods of the current study are incompletely described and referenced, which call the validity of the results and conclusions into question. Please provide the necessary level of detail in the methods so that the study may be reproducible by others. The Discussion should also address the limitations of the data analysis more clearly and acknowledge how this study advances knowledge in the context of other recent and relevant background literature highlighted by the reviewers.

Reviewer's Responses to Questions

**Key Review Criteria Required for Acceptance?**

**Methods**

-Are the objectives of the study clearly articulated with a clear testable hypothesis stated?

-Is the study design appropriate to address the stated objectives?

-Is the population clearly described and appropriate for the hypothesis being tested?

-Is the sample size sufficient to ensure adequate power to address the hypothesis being tested?

-Were correct statistical analysis used to support conclusions?

-Are there concerns about ethical or regulatory requirements being met?

Reviewer #1: Line 122 – please include some brief information about the services available at the health facilities – was rabies immunoglobulin available in any regions during the study period? Were human rabies vaccines widely available throughout the study period or were there shortages? Please describe the standard PEP regimen used in Uganda during the study period. Is there any data available for the total number of doses received? 

Line 122 - If the veterinary/ AH officer issues a referral letter for PEP, but then that bite victim does not attend the health facility, or there is no vaccine/RIG available when they arrive there, how is this recorded in the data? If this still gets recorded as ‘received PEP’, then rates of rPEP uptake may be overestimated. This should be clearly stated in the methods, but also as a limitation when interpreting the results around PEP compliance.

Line 124 – Is it possible to include the standard surveillance form as Supplementary Materials?

Line 128 – It is not clear how the ‘survival status of the bitten human’ is reported. The data for the study is described as being gathered by the veterinary officer/animal husbandry officer at the time of the bite and then reported through the DVO and CAH to NADDEC. As any human death would occur weeks or months after the bite, how is this data gathered and reported? Although some information is provided in the Limitations (Line 255 – 256), please state in the methods whether the human rabies data is for suspect human cases based solely on clinical presentation or confirmed by laboratory diagnosis. Is there a system of reporting suspect human rabies cases by hospitals back to the veterinary or AH officer who then reported it through the DVO to CAH to NADDEC or was this data reported to NADDEC by the Department of Health? Please clarify the methods around this data.

Line 134 – Please state what the key variables were. For example I assume if the vaccination status of the animal were missing, the record was not excluded.

Reviewer #2: The objectives and population of the study are clearly written in the methods section with plain language. The authors present a simple analysis of national level surveillance data from a veterinary public health reporting system. Some specific areas for revision include the following:

Line 134: What are the key variables? Indicate completeness of data in the results in addition to overall reporting rates

Line 136: How are the number of expected reports calculated?

Line147: How does total human population at risk differ from total population?

Reviewer #3: My main concern regarding the present manuscript is in the methods section, which is poorly described and I missing a lot of information. The authors need to provide details on how the study was designed, and the statistical methods used.

**Results**

-Does the analysis presented match the analysis plan?

-Are the results clearly and completely presented?

-Are the figures (Tables, Images) of sufficient quality for clarity?

Reviewer #1: 164 – 165 – Please include how the incidence of rabies in animals change during the study period. It would also be beneficial to plot the geographic animal rabies incidence by year on Figure 2, perhaps as a bubble plot on top of the choropleth map of bite incidence.

Line 167 – Was the number of doses of rPEP recorded, or only that they received the first dose?

Line 169 – 170 – Is there correlation between the geographic and temporal distribution of animal rabies cases and that of human rabies? Again some information about the geographic incidence of human rabies cases, like for animals, would be important to include.

Line 169 – 171 – Please also include the number of human deaths with the percentages.

Line 175 – The estimate of 2.2 million owned dogs should be compared with the analysis published by Wallace et al (2017), which estimated 730,000 owned dogs in Uganda. 

Line 175 - The use of population estimates is not recommended as a basis for estimating vaccination coverage (Sambo et al 2017), however the point that insufficient dog vaccinations are taking place in Uganda is valid. A national vaccination coverage is also very misleading as it gives no indication of the heterogeneity of vaccination effort across the country. Intensive regions of vaccination are most important in the early stages to develop experience and methods before expanding nationally, however this would not be clear from only reporting a nationally low vaccination coverage. A recent paper (Evans et al 2019) reported high vaccination coverage through a static point vaccination approach in Nwoya District.

Figure 1 – This figure is generally cumbersome for the reader to interpret, moving between two axis, the colour legend and the line graphs, as well as percentages with different denominators. It is also difficult to make any interpretation of mortality rate which is too small on the scale and has no reference to what this proportion is relating to (i.e. percentage mortality in bite victims). I think this graph would therefore be more effective and impactful if split into at least two graphs. Perhaps a staked bar graph with total number of animal bite cases (y-axis) by year (x-axis), with bars coloured to show the number of cases within these that received rPEP. Human mortality rate could also be shown on this graph as a line graph on a second y-axis of total human deaths by year. I think reporting rate should be displayed on a second figure with additional explanation about what this relates to e.g. Percentage of reports received from district as compared to the expected number of reports.

REFERENCES

Wallace, R.; Mehal, J.; Y, N.; S, R.; B, B.; M, O.; V, T.; JD, B.; A, G.; J, W. The impact of poverty on dog ownership and access to canine rabies vaccination: Results from a knowledge, attitudes and practices survey, Uganda 2013. 2017, 1–22.

Sambo, M.; Johnson, P.C.D.; Hotopp, K.; Changalucha, J. Comparing Methods of Assessing Dog Rabies Vaccination Coverage in Rural and Urban Communities in Tanzania. 2017, 4.

Evans, M.J.; Burdon Bailey, J.L.; Lohr, F.E.; Opira, W.; Migadde, M.; Gibson, A.D.; Handel, I.G.; Bronsvoort, B.M. d. C.; Mellanby, R.J.; Gamble, L.; et al. Implementation of high coverage mass rabies vaccination in rural Uganda using predominantly static point methodology. Vet. J. 2019, 249, 60–66.

Reviewer #2: Results from the analysis are presented clearly. Figures and legends could use some additional information for clarity. There are some concerns for the completeness (only 38% of jurisdictions reported, bite events might not be linked to fatal rabies outcomes) and correctness (incidence was calculated from unverified dog population and outdated census information). Many of the trends reported in the results are likely influenced by a change in reporting over the five year period. Specific areas for revision include:

Would suggest breaking down the number of pets into dogs and cats specifically

Figure 2: Do these administrative divisions represent districts or regions? I suggest the legend be updated to include this information. Since regions are mentioned in the text it would be helpful to the reader if the regional borders are shown here. May be helpful to include a layer for water (lake Victoria) and large cities

Line 167: Clarify if receiving rPEP mean the individual completed the entire 4/5 dose series?

Line 171-173: Where does the 2.24M pets number come from? Is that the entire pet population of Uganda? Please clarify how the 62% of dogs destroyed but communities was calculated

Conclusions:

Line 237-238: Stating that because only a few of he persons bitten by animals died could be misinterpreted by readers that rPEP should not we a concern after a bite

Line 247-250: given the low levels of dog vaccination presented here there are likely numerous endemic factors that could lead to higher rabies rates in addition to only importation

Reviewer #3: My main concerns in the results section is how the statistical analysis was performed. I do not know if the authors used appropriated statistical tests.

**Conclusions**

-Are the conclusions supported by the data presented?

-Are the limitations of analysis clearly described?

-Do the authors discuss how these data can be helpful to advance our understanding of the topic under study?

-Is public health relevance addressed?

Reviewer #1: Line 181 – 183 – I think these headline conclusions need to be given more context from the outset as it seems the decline in bite cases could be explained by the decline in reporting during the study period as opposed to a true decline in incidence. It would be important to also give the human mortality data more context; perhaps that there was a decline in the proportion of bite victims which were reported to have died of rabies (depending on how the human rabies death data was gathered).

Line 183 – 184 – The fact that most of the animal samples did not test positive for rabies does not give us any information about the rabies situation. Higher rates of negative results are expected as mentioned in the discussion. It would be more relevant to report any change in animal rabies incidence over time.

Line 198 – 200 – Perhaps the use of eHealth and mobile technology could improve the efficiency and timeliness of case reporting as has been demonstrated elsewhere without increasing cost? (Mtema et al 2016, Gibson et al 2018)

Line 206 – 208 – The reference cited here (Lembo et al 2010) is a review which highlights that most dogs are accessible for vaccination as opposed to this being a major limitation. Several studies show that a sufficient proportion of dogs in most African communities are amenable to handling for parenteral vaccination. The Wallace et al (2017) study indicates that the total dog population in Uganda may be less than previously estimated and Evans et al (2019) indicate that over 80% of dogs could be vaccinated in a pilot campaign through Static Point vaccination. These studies should be included in the discussion of the results of this paper and help to consider the prospects for rabies control in Uganda.

Line 224 – 232 – Methods for Integrated Bite Case Management (IBCM) have shown considerable benefits in improving the investigation of suspect rabies cases (Etheart 2017, Medley 2017). It would be worth considering in the discussion if these could also be of benefit in Uganda to increase compliance with PEP in cases of highest risk.

Line 254 – 256 – Here it states that it was not possible to know if the person who died did or did not receive rPEP, however in Line 237 – 238 it states that few people of those who did not receive PEP, few died. This seems contradictory if it is known who of the people who did not receive PEP died. This stems from the lack of clarity around how the human rabies incidence data was obtained. Please clarify.

REFERENCES

Mtema, Z.; Changalucha, J.; Cleaveland, S.; Elias, M.; Ferguson, M.; Halliday, J.E.B.; Haydon, D.T.; Jaswant, G. Mobile Phones As Surveillance Tools : Implementing and Evaluating a Large-Scale Intersectoral Surveillance System for Rabies in Tanzania. PLoS Negl. Trop. Dis. 2016, 1–12.

Gibson, A.D.; Mazeri, S.; Lohr, F.; Mayer, D.; Burdon, J.L.; Wallace, R.M.; Handel, I.G.; Shervell, K.; Bronsvoort, B.M.; Mellanby, R.J.; et al. One million dog vaccinations recorded on mHealth innovation used to direct teams in numerous rabies control campaigns. PLoS One 2018, 13.

Evans, M.J.; Burdon Bailey, J.L.; Lohr, F.E.; Opira, W.; Migadde, M.; Gibson, A.D.; Handel, I.G.; Bronsvoort, B.M. d. C.; Mellanby, R.J.; Gamble, L.; et al. Implementation of high coverage mass rabies vaccination in rural Uganda using predominantly static point methodology. Vet. J. 2019, 249, 60–66.

Etheart, M.D.; Kligerman, M.; Augustin, P.D.; Blanton, J.D.; Monroe, B.; Fleurinord, L.; Millien, M.; Crowdis, K.; Fenelon, N.; Wallace, R.M. Effect of counselling on health-care-seeking behaviours and rabies vaccination adherence after dog bites in Haiti , 2014 – 15 : a retrospective follow-up survey. Lancet Glob. Heal. 2017, 5, e1017–e1025.

Medley, A.; Millien, M.; Blanton, J.; Ma, X.; Augustin, P.; Crowdis, K.; Wallace, R. Retrospective Cohort Study to Assess the Risk of Rabies in Biting Dogs, 2013-2015, Republic of Haiti. Trop. Med. Infect. Dis. 2017, 2, 14.

Reviewer #2: The conclusions presented by the author are not fully supported by the data. While they do note several limitations, they are understated and could lead to the results of the manuscript being taken out of context. The authors do present reasons why this type of analysis is important in Uganda and for others working in public health in low resource settings. Specific comments include:

Line 237-238: Stating that because only a few of he persons bitten by animals died could be misinterpreted by readers that rPEP should not we a concern after a bite

Line 247-250: given the low levels of dog vaccination presented here there are likely numerous endemic factors that could lead to higher rabies rates in addition to only importation

Reviewer #3: (No Response)

**Editorial and Data Presentation Modifications?**

Reviewer #1: Line 59 – amend to ‘rabies causes an estimated 59,000 deaths’ as opposed to ‘at least’.

Line 68 – 73 – Long sentence, suggest splitting in two for readability.

Reference formatting needs to be reviewed throughout the References section. Some examples include References 9, 11, 12, 17, 18, 26, 29, 30.

Punctuation should be reviewed throughout. E.g. Line 148, Line 153.

Reviewer #2: Minor revisions. Suggest the authors review the text and change some terms to be more in line with the scientific literature (e.g. destroying dogs to euthanasia). Some references are not peer-reviewed

Reviewer #3: Major revision

**Summary and General Comments**

Reviewer #1: Congratulations to the authors on this relevant study which describes the current rabies surveillance activities in Uganda. The manuscript is well written and the results are of relevance to continuing to develop comprehensive rabies surveillance systems in Uganda and other East African countries. Inclusion of the temporal and geographic distribution of both human and canine rabies incidence is crucial to the study and to planning a national dog vaccination initiative. This data would be of benefit as it would be logical and cost-effective to first scale intensive dog vaccination efforts in districts with the greatest human burden, generating effective methods and demonstrating examples of success to apply elsewhere in the country. 

The source of human rabies incidence data is not clear in the current manuscript which makes interpretation of these results difficult. The lack of information around whether bite victims received a single dose or full course of rPEP and the potential for over-reporting rPEP uptake as described in comments is a concern for interpreting the results. It would be of great concen if a high proportion of bite victims who have been exposed to a rabid dog only take a single dose of rPEP, whereas compliance in victims who were bitten by vaccinated, healthy pet dogs would be of less concern. The use of basic IBCM protocols could help to generate a more complete picture of rabies exposures and PEP compliance in different risk groups. This appears to be the top priority for reducing risk to rabies through existing health care resources.

Reviewer #2: The authors present a summary of veterinary surveillance data for animal bites and rabies prevention efforts in Uganda for a 5-year time period. The data includes both periods of active and passive surveillance in the national system that is marked by a significant decrease in reporting. Data for incidence and proportion used national census data for both humans and animals. The data, which are largely incomplete, show a related downward trend in both animal bite incidence and reporting. The authors also summarize animal specimens tested for rabies, persons receiving post exposure prophylaxis, and community culling of aggressive dogs. 

The manuscript was well-written using plain language. It provides new information from systematically collected data from an area of the world where it is difficult to come by. The content will be useful to public health officials and researchers who work on rabies in Africa and other low-resource setting. 

The limitations of the data used in this study are presented in the results and discussion sections but may not be properly emphasized. This could lead to the publication being improperly cited. I would suggest the authors provide more detail.

Reviewer #3: In the manuscript entitled “Trends and spatial of animal bites and vaccination status among victims and the animal population, Uganda: A veterinary surveillance system analysis, 2013–2017”, the authors describe important results regarding incidence and spatial distribution of animal bites, PEP use, and mortality rates due to rabies disease in Uganda. However, the manuscript is missing important and precise information on the methods and a weak discussion and conclusion. Please, see my specific comments below.

Introduction

Line 58: All mammals? Is it not too much?

Line 64: I recommend changing the word “Almost” for something like…

Lines 65-73: I don’t think is necessary to describe the clinical course of the disease in dogs and humans. I recommend suggest deleting the entire paragraph.

Line 75: “Prevention of rabies in animals is primarily achieved through vaccination”. This is the focus of your manuscript.

Lines 92-93: Please clarify why the cost of rabies vaccine is “prohibitive”.

Lines 95-96: This first sentence looks confuse. Please rewrite to have a better flow, or I recommend deleting this sentence.

Lines 101-102: “However, Uganda also captures rabies-related data in a veterinary surveillance system, at National Animal Disease Diagnostic Epidemiology Centre (NADDEC)”. I think this sentence seems disconnected. However, Uganda also captures rabies-related data, and…? I recommend merging with the next sentence to have a better flow.

Line 103: Please clarify “mandated”.

Major: The introduction section is too long and makes it lose the main focus of the manuscript. My suggestion is to delete some excessive information and focus on what your study brings of most important.

Methods

Lines 117-118: “The officer visits the scene of the incident to ensure that the animal (usually dog/cat) is killed and its head packaged for shipment to NADDEC for analysis”. The office always kills the SUSPECTED animals? Is there any observation at the first moment? Please clarify.

Lines 126-131: I recommend creating another topic to explain the variables were capture for further analysis, giving more details and adding information about inclusion and exclusion criteria.

Lines 139-141: Same here, please provide further details about using laboratory records.

Lines 145-146: “Regional and national human population data were obtained from the Uganda National Census 2014” and? What other information did the authors collect? Please provide details.

Major: The methods section is poorly described and I missed a lot of information. The authors need to provide details on how the study was designed, how the data was collected specifically for each group they want to show results (dogs, humans, distribution maps, PEP data, mortality data, etc). The data analysis section is confused. The authors should separate acquiring data from analyzing data. What statistical methods were used and why? I also recommend a general map of Uganda to show the readers the features of the country and how it looks like in terms of geography and how population (dogs and humans) are distributed

Results

Lines 156-159: Great, but how did the authors calculate this?

Lines 160-163: Please provide statistical data to support the differences among regions (although it’s clear in the figure).

Lines 171-174: What does it mean “6,997 pets were reported destroyed by the communities”? Please clarify.

Discussion

Lines 187-192: I think authors should also include here a discussion about the increase of animal bites in 2014.

Lines 202-204: “Vaccinating pets…” Only pets? Please clarify.

Other comments:

Did the authors separate bite rates from dogs, cats, or wild animals?

8420 persons reported animal bites. How many from dogs, cats, or wild?

Western region had a higher incidence. Can you add risk ratio or any other statistics?

Any recommendations regarding education of the population?

PLOS authors have the option to publish the peer review history of their article (what does this mean?). If published, this will include your full peer review and any attached files.

Reviewer #1: No

Reviewer #2: Yes: Benjamin P Monroe

Reviewer #3: Yes: Galileu Barbosa Costa
---

## [Decision Letter · Decision Letter 1]

10 Aug 2020

Dear Dr Monje,

Thank you very much for submitting your manuscript "Trends and spatial distribution of animal bites and vaccination status among victims and the animal population, Uganda: A veterinary surveillance system analysis, 2013 – 2017" for consideration at PLOS Neglected Tropical Diseases. As with all papers reviewed by the journal, your manuscript was reviewed by members of the editorial board and by several independent reviewers. The reviewers appreciated the attention to an important topic. Based on the reviews, we are likely to accept this manuscript for publication, providing that you modify the manuscript according to the review recommendations. 

Reviewers agree that the manuscript is much improved, but there are still some minor suggestions that need to be addressed below

Sincerely,

Daniel Leo Horton, PhD

Associate Editor

Sergio Recuenco

Deputy Editor

Reviewers agree that the manuscript is much improved, but there are still some minor suggestions that need to be addressed below

Reviewer's Responses to Questions

**Key Review Criteria Required for Acceptance?**

**Methods**

-Are the objectives of the study clearly articulated with a clear testable hypothesis stated?

-Is the study design appropriate to address the stated objectives?

-Is the population clearly described and appropriate for the hypothesis being tested?

-Is the sample size sufficient to ensure adequate power to address the hypothesis being tested?

-Were correct statistical analysis used to support conclusions?

-Are there concerns about ethical or regulatory requirements being met?

Reviewer #1: (No Response)

Reviewer #2: The objectives of the study are clear and the methods used are standard for this type of analysis. The manuscript is evaluating a public health surveillance system and no ethical or regulatory concerns are apparent.

**Results**

-Does the analysis presented match the analysis plan?

-Are the results clearly and completely presented?

-Are the figures (Tables, Images) of sufficient quality for clarity?

Reviewer #1: (No Response)

Reviewer #2: The results match the proposed analysis and have been clarified over the previous version. Figures are adequate for this type of information

What is the total population of Uganda that was used to calculate incidence?

How many records were removed for unknown district or date?

line 179: suggest authors replace 'received' with 'initiated'

**Conclusions**

-Are the conclusions supported by the data presented?

-Are the limitations of analysis clearly described?

-Do the authors discuss how these data can be helpful to advance our understanding of the topic under study?

-Is public health relevance addressed?

Reviewer #1: (No Response)

Reviewer #2: The public health relevance of this topic is important and the authors provide some generalizations from their analysis that highlight the issues with rabies surveillance in Afria. They highlight how changing from active to passive surveillance can influence reporting rates and by extension incidence. While the limitations of the analysis are presented, I would caution that some of their conclusions about overall rabies risk in Uganda could be more nuanced.

**Editorial and Data Presentation Modifications?**

Reviewer #1: (No Response)

Reviewer #2: The current version clarifies many of the issues of the original version. Some minor editing of language and punctuation my improve the overall readability of the manuscript.

**Summary and General Comments**

Reviewer #1: Thank you to the authors for addressing the reviewer comments, which has improved the clarity of the study, particularly in the methods. I feel the discussion and conclusions have been given additional context and the limitations of the study have been more clearly stated. The study still presents data which will be useful in planning rabies control activities and identifies challenges in progressing disease control efforts in the region. I have a few minor comments which would need to be addressed before the manuscript would be suitable for publication.

In the abstract and elsewhere in the manuscript, greater clarity around what ‘reporting’ refers to is needed. Specifically the distinction between ‘decreased reporting’ of dog bites, i.e. a reduction in people reporting to healthcare facilities with dog bites, as compared to decreased reporting of bite data through district-level human or veterinary government reporting channels (Line 34, Line 35, Line 198 – 199, Line 290).

Line 193-195 - The first paragraph in the discussion remains misleading; “…shows an overall decline in animal bites in Uganda during 2013 – 2017”. This does not appear to be the case from the data in this study, but rather the overall animal bite rate in Uganda cannot be determined due to changes in the regularity of district-level reporting of data.

Line 222 – 224 - The reason for dismissal of the previously published peer-reviewed study of dog population enumeration in Uganda is not valid from a sample size perspective. Although a relatively small fraction of regions were sampled in this study, the methods appear statistically sound and the power of the sample size adequate, which would have enabled valid estimation of the national population. The initial estimate of the study at 1.3 million dogs is not hugely dissimilar to that of the livestock census at 1.6 million dogs. It is fair to comment on the differences between these estimates and perhaps to consider the Livestock census the more valid for purposes of this study, but the comment of sample size is not be valid from a statistical sense. Ultimately whether the total dog population is estimated at 0.7 million, 1.3 million or 1.6 million, the conclusion remains the same; dog vaccination in Uganda is inadequate and a systematic national dog vaccination campaign is needed. 

Figure 1 – This figure still lacks clarity and is difficult to interpret. It is not immediately clear which axis refers to which line graph and the ‘Mortality rate in bite victims’ is not clear in the current scale. This may be more clearly displayed in stacked graphs aligned by year on the x-axis, but with distinct y-axes scales and labels. This could be presented as a single figure.

Reviewer #2: The manuscript provides important information for persons interested in canine rabies elimination. It highlights many of the issues faced by government institutions in low resource settings including incomplete reporting, unreliable population data, laboratory testing capacity, and inability to follow patient outcomes.

PLOS authors have the option to publish the peer review history of their article (what does this mean?). If published, this will include your full peer review and any attached files.

Reviewer #1: No

Reviewer #2: No
---

## [Decision Letter · Decision Letter 2]

8 Dec 2020

Dear Dr Monje,

Thank you very much for submitting your manuscript "Trends and spatial distribution of animal bites and vaccination status among victims and the animal population, Uganda: A veterinary surveillance system analysis, 2013 – 2017" for consideration at PLOS Neglected Tropical Diseases. As with all papers reviewed by the journal, your manuscript was reviewed by members of the editorial board and by several independent reviewers. The reviewers appreciated the attention to an important topic. Based on the reviews, we are likely to accept this manuscript for publication, providing that you modify the manuscript according to the review recommendations. 

Thank you for responding to the reviewers comments. Reviewer 1 has made further suggestions just to Figure 1, which I agree would enhance the manuscript further. If you decide not to make these changes please provide justification.

Sincerely,

Daniel Leo Horton, PhD

Associate Editor

Sergio Recuenco

Deputy Editor

Thank you for responding to the reviewers comments. Reviewer 1 has made further suggestions just to Figure 1, which I agree would enhance the manuscript further. If you decide not to make these changes please provide justification.

Reviewer's Responses to Questions

**Key Review Criteria Required for Acceptance?**

**Methods**

-Are the objectives of the study clearly articulated with a clear testable hypothesis stated?

-Is the study design appropriate to address the stated objectives?

-Is the population clearly described and appropriate for the hypothesis being tested?

-Is the sample size sufficient to ensure adequate power to address the hypothesis being tested?

-Were correct statistical analysis used to support conclusions?

-Are there concerns about ethical or regulatory requirements being met?

Reviewer #1: (No Response)

**Results**

-Does the analysis presented match the analysis plan?

-Are the results clearly and completely presented?

-Are the figures (Tables, Images) of sufficient quality for clarity?

Reviewer #1: (No Response)

**Conclusions**

-Are the conclusions supported by the data presented?

-Are the limitations of analysis clearly described?

-Do the authors discuss how these data can be helpful to advance our understanding of the topic under study?

-Is public health relevance addressed?

Reviewer #1: (No Response)

**Editorial and Data Presentation Modifications?**

Reviewer #1: (No Response)

**Summary and General Comments**

Reviewer #1: Thank you to the authors for their thorough revisions in response to previous comments, which have improved the clarity of conclusions drawn from the results. All concerns in the text have been addressed, however some final adjustment to Figure 1 would considerably benefit the clarity of communicating key points in the data. 

I'm sorry for poorly communicating the suggestion of a stacked graph in previous comments. As opposed to stacked bar graphs, I was trying to suggest stacked line graphs with each plotted on an independent y-axis. This avoids the need for additional labelling / colour which can hamper interpretation. Each axis has it's own scale and label as opposed to grouping multiple variables into one axis, allowing the reader to immediately understand what data is being shown in each line. An example of this approach can be seen here: http://www.performance-ideas.com/2012/03/27/stacked-line-charts/

Other than this concern the manuscript appears suitable for publication and would be of interest to the readership of PLOS NTD.

PLOS authors have the option to publish the peer review history of their article (what does this mean?). If published, this will include your full peer review and any attached files.

Reviewer #1: No
---

## [Editor Report · Decision Letter 3]

6 Jan 2021

Dear Dr Monje,

We are pleased to inform you that your manuscript 'Trends and spatial distribution of animal bites and vaccination status among victims and the animal population, Uganda: A veterinary surveillance system analysis, 2013 – 2017' has been provisionally accepted for publication in PLOS Neglected Tropical Diseases.

Best regards,

Daniel Leo Horton, PhD

Associate Editor

Sergio Recuenco

Deputy Editor

---

## [Editor Report · Acceptance letter]

7 Apr 2021

Dear Dr Monje,

We are delighted to inform you that your manuscript, "Trends and spatial distribution of animal bites and vaccination status among victims and the animal population, Uganda: A veterinary surveillance system analysis, 2013 – 2017," has been formally accepted for publication in PLOS Neglected Tropical Diseases.

Best regards,

Shaden Kamhawi

co-Editor-in-Chief

Paul Brindley

co-Editor-in-Chief
